# GRAPH MIXUP WITH SOFT ALIGNMENTS

## ABSTRACT

We study graph data augmentation by mixup, which has been used successfully on images. A key operation of mixup is to compute a convex combination of a pair of inputs. This operation is straightforward for grid-like data, such as images, but challenging for graph data. The key difficulty lies in the fact that different graphs typically have different numbers of nodes, and thus there lacks a node-level correspondence between graphs. In this work, we propose a simple yet effective mixup method for graph classification by soft alignments. Specifically, given a pair of graphs, we explicitly obtain node-level correspondence via computing a soft assignment matrix to match the nodes between two graphs. Based on the soft assignments, we transform the adjacency and node feature matrices of one graph, so that the transformed graph is aligned with the other graph. In this way, any pair of graphs can be mixed directly to generate an augmented graph. We conduct systematic experiments to show that our method can improve the performance and generalization of graph neural networks (GNNs) on various graph classification tasks. In addition, we show that our method can increase the robustness of GNNs against noisy labels.

## 1 INTRODUCTION

Data augmentations aim at generating new training samples by applying certain transformations on the original samples. For example, applying rotations and flipping on images generates new images with the same labels. Many empirical results have shown that data augmentations can help improve the invariance and thus the generalization abilities of deep learning models. While data augmentations are relatively straightforward for grid-like data, such as images, they are particularly challenging for graph data. A key difficulty lies in the lack of simple graph operations that preserve the original labels, such as rotations on images. Most existing graph augmentation methods, such as DropEdge (Rong et al., 2019), DropNode (Feng et al., 2020) and Subgraph (You et al., 2020), assume labels are the same after simple operations, such as drop a random node or edge, on training graphs. On one hand, such simple operations may not be able to generate sufficiently diverse new samples. On the other hand, although the operations are simple, they are not guaranteed to preserve the original labels.

Recently, mixup (Zhang et al., 2017) has been shown to be an effective method for image data augmentation. In particular, mixup generates new samples and corresponding labels by performing convex combinations of a pair of original samples and labels. A key challenge of applying mixup on graphs lies in the fact that different graphs typically have different numbers of nodes. Even for graphs with the same number of nodes, there lacks a node-level correspondence that is required to perform mixup. Several existing graph mixup methods (Han et al., 2022; Park et al., 2022; Yoo et al., 2022; Guo & Mao, 2021) use various tricks to sidestep this problem. For example, ifMixup (Guo & Mao, 2021) uses a random node order to align graphs and then interpolate node feature matrices and adjacency matrices. Han et al. (2022) proposes to learn a Graphon for each class and performs mixup in Graphon space. Graph Transplant (Park et al., 2022) and SubMix (Yoo et al., 2022) connect subgraphs from different input graphs to generate new graphs. However, none of these methods explicitly models the node-level correspondence among different graphs and perform mixup as in the case of images. A natural question is raised: *Can we conduct image-like mixup for graphs with node-level correspondence to preserve critical information?*

In this work, we provide an affirmative answer to this question and propose a simple yet effective graph mixup approach via soft alignments. A key design principle of our method is to explicitly

and automatically model the node-level correspondence (i.e., soft alignment matrix) between two graphs when performing mixup, thereby avoiding random matching noise and preserving critical graph components in the augmented data. Given a pair of graphs, we first obtain node-level correspondence by computing a soft assignment matrix that measures the similarity of nodes across two graphs based on node features and graph topology. Then this soft alignment matrix guides the graph transformation, including adjacency matrix and node feature matrix transformation, to generate the aligned graph with the same number of nodes and node order as the other graph. In this way, we can interpolate the adjacency matrices and node feature matrices of any graph pairs to generate synthetic graphs for training. We conduct comprehensive experiments to evaluate our method. Results show that our method can improve the performance and generalization of GNNs on various graph classification tasks. In addition, results show that our method increases the robustness of GNNs against noisy labels.

## 2 PRELIMINARIES

### 2.1 GRAPH CLASSIFICATION WITH GRAPH NEURAL NETWORKS

In this work, we study the problem of graph classification. Let $\mathcal{G} = (\boldsymbol{A}, \boldsymbol{X})$ represent a graph with $n$ nodes. Here, $\boldsymbol{A} \in \{0, 1\}^{n \times n}$ is the adjacency matrix, and $\boldsymbol{A}_{i,j} = 1$ if and only if there exists an edge between nodes $i$ and $j$. $\boldsymbol{X} = [\boldsymbol{x}_1, \cdots, \boldsymbol{x}_n]^T \in \mathbb{R}^{n \times d}$ is the node feature matrix, where each row $\boldsymbol{x}_i \in \mathbb{R}^d$ represents the $d$-dimensional feature of node $i$. Given a set of labeled graphs, graph classification tasks aim to learn a model that predicts the class label $y$ of each graph $\mathcal{G}$. Recently, GNNs have shown remarkable performance in various graph classification problems. GNNs usually use a message passing scheme to learn node representations in graphs. Let $\boldsymbol{H}^{(l)} = [\boldsymbol{h}_1^{(l)}, \cdots, \boldsymbol{h}_n^{(l)}]^T \in \mathbb{R}^{n \times d_l}$ denote the node representations at the $l$-th layer of a message passing GNN model, where each row $\boldsymbol{h}_i^{(l)} \in \mathbb{R}^{d_l}$ is the $d_l$-dimensional representation of node $i$. Formally, one message passing layer can be described as

$$\boldsymbol{H}^{(l)} = \text{UPDATE}(\boldsymbol{H}^{(l-1)}, \text{MSG}(\boldsymbol{H}^{(l-1)}, \boldsymbol{A})), \tag{1}$$

where $\text{MSG}(\cdot)$ is a message propagation function that aggregates the messages from neighbors of each node, and $\text{UPDATE}(\cdot)$ is a function that updates $\boldsymbol{H}^{(l-1)}$ to $\boldsymbol{H}^{(l)}$ using the aggregated messages. The node representations $\boldsymbol{H}^{(0)}$ are initialized as $\boldsymbol{X}$. After $L$ layers of such message passing, the graph representation $\boldsymbol{h}_{\mathcal{G}}$ is obtained by applying a global pooling function READOUT over node representations as

$$\boldsymbol{h}_{\mathcal{G}} = \text{READOUT}(\boldsymbol{H}^{(L)}). \tag{2}$$

Given the graph representation $\boldsymbol{h}_{\mathcal{G}}$, a multi-layer perceptron (MLP) model computes the probability that graph $\mathcal{G}$ belongs to each class.

Despite the success of GNNs, a primary challenge in graph classification tasks is the lack of labeled data due to expensive annotations. In this paper, we focus on designing a pairwise graph data augmentation method to generate more training data, thereby improving the performance of GNNs.

### 2.2 MIXUP

Mixup (Zhang et al., 2017) is a data augmentation method for regular, grid-like, and Euclidean data such as images and tabular data. The idea of mixup is to linearly interpolate random pairs of data samples and their corresponding labels. Given a random pair of samples $\boldsymbol{x}_i$ and $\boldsymbol{x}_j$ and their corresponding one-hot class labels $\boldsymbol{y}_i$ and $\boldsymbol{y}_j$, Mixup constructs training data as

$$\tilde{\boldsymbol{x}} = \lambda \boldsymbol{x}_i + (1 - \lambda) \boldsymbol{x}_j, \quad \tilde{\boldsymbol{y}} = \lambda \boldsymbol{y}_i + (1 - \lambda) \boldsymbol{y}_j, \tag{3}$$

where $\lambda \sim \text{Beta}(\alpha, \alpha)$ is a random variable drawn from the Beta distribution parameterized with $\alpha$.

Mixup and its variants (Yang et al., 2020; Yun et al., 2019; Berthelot et al., 2019) have shown great success in improving the generalization and robustness of deep neural networks in image recognition and natural language processing. However, mixing graphs is a challenging problem due to the irregular and non-Euclidean structure of graph data. Specifically, the number of nodes varies in different graphs, making it infeasible to apply the mixing rule in Eq. (3) directly. Even if two

graphs have the same number of nodes, graphs have no inherent node order. If we don't consider the node-level correspondence between graphs and use an arbitrary node order to mix graphs, the generated graphs are noisy.

## 3 METHODOLOGY

In this work, we propose S-Mixup, a novel and effective mixup method for graph classification, which addresses the challenges of mixing graph data. We compute a soft assignment matrix to match the nodes between a pair of graphs. The assignment matrix guides the graph transformation to well align graph pairs based on node attributes and graph topology, such that the augmented new graph can preserve the critical information and avoid random matching noise.

### 3.1 MIXUP WITH AN ASSIGNMENT MATRIX

Assuming that we already have the desired soft assignment matrix, we first describe how we mix graphs based on a soft assignment matrix. Given a pair of graphs $\mathcal{G}_1 = (\boldsymbol{A}_1, \boldsymbol{X}_1)$ and $\mathcal{G}_2 = (\boldsymbol{A}_2, \boldsymbol{X}_2)$, we use $\boldsymbol{M} \in \mathbb{R}^{n_1 \times n_2}$ to denote the soft assignment matrix, where $n_1$ and $n_2$ are the number of nodes in $\mathcal{G}_1$ and $\mathcal{G}_2$, respectively. Each row of $M$ corresponds to a node in $\mathcal{G}_1$ and each column of $M$ corresponds to a node in $\mathcal{G}_2$. The soft assignment matrix $\boldsymbol{M}$ represents the node-level correspondence between $\mathcal{G}_1$ and $\mathcal{G}_2$. In other words, the entry $\boldsymbol{M}_{i,j}$ denotes the likeness that the node $j$ in $\mathcal{G}_2$ is matched to the node $i$ in $\mathcal{G}_1$.

Given the assignment matrix $\boldsymbol{M}$, we transform $\mathcal{G}_2 = (\boldsymbol{A}_2, \boldsymbol{X}_2)$ to $\mathcal{G}_2' = (\boldsymbol{A}_2', \boldsymbol{X}_2')$ as

$$\boldsymbol{A}_2' = \boldsymbol{M}\boldsymbol{A}_2\boldsymbol{M}^T, \quad \boldsymbol{X}_2' = \boldsymbol{M}\boldsymbol{X}_2. \tag{4}$$

After transformation, $\mathcal{G}_2'$ is aligned well with $\mathcal{G}_1$ via a node-level one-to-one mapping. In this way, we can generate a new graph $\mathcal{G}' = (\boldsymbol{A}', \boldsymbol{X}')$ via mixing up graphs $\mathcal{G}_1$ and $\mathcal{G}_2'$. To be specific, $\mathcal{G}'$ is generated via linear interpolation on both node features and topological structures. Formally, this process can be described as

$$\begin{aligned} \boldsymbol{X}' &= \lambda\boldsymbol{X}_1 + (1-\lambda)\boldsymbol{M}\boldsymbol{X}_2, \\ \boldsymbol{A}' &= \lambda\boldsymbol{A}_1 + (1-\lambda)\boldsymbol{M}\boldsymbol{A}_2\boldsymbol{M}^T, \\ \boldsymbol{y}' &= \lambda\boldsymbol{y}_1 + (1-\lambda)\boldsymbol{y}_2, \end{aligned} \tag{5}$$

where $\boldsymbol{y}_1$ and $\boldsymbol{y}_2$ are the one-hot class labels of graph $\mathcal{G}_1$ and $\mathcal{G}_2$, respectively. The mixup ratio $\lambda = \max(\lambda', 1 - \lambda')$, where $\lambda' \in [0, 1]$ is sampled from a $\text{Beta}(\alpha, \alpha)$ distribution with a hyperparameter $\alpha$. Note that the generated new graph $\mathcal{G}' = (\boldsymbol{A}', \boldsymbol{X}')$ is a fully connected edge-weighted graph. In other words, $\boldsymbol{A}' \in [0, 1]^{n_1 \times n_1}$ is a real matrix, where each entry $\boldsymbol{A}'_{i,j}$ denotes the weight of the edge between nodes $i$ and $j$ in $\mathcal{G}'$. Together with the label $\boldsymbol{y}'$, the generated new graph $\mathcal{G}'$ is used as the augmented training data.

### 3.2 COMPUTING THE ASSIGNMENT MATRIX

Since we need to perform soft alignments for all input graph pairs, an accurate assignment matrix with efficient computation is in need. Thereby, we propose to compute the node-level assignment matrix based on a graph matching network (Li et al., 2019), which is used to compare graph-level similarities. Specifically, a pair of graphs $\mathcal{G}_1$ and $\mathcal{G}_2$ is taken as input to extract a pair of node representations $\boldsymbol{H}_1$ and $\boldsymbol{H}_2$ by message passing within and between graphs. Formally, the message passing process of node representations $\boldsymbol{H}_1^{(l)}$ in $\mathcal{G}_1$ at $l$-th layer can be formulated as

$$\boldsymbol{H}_1^{(l)} = \text{UPDATE}(\boldsymbol{H}_1^{(l-1)}, \text{MSG}_1(\boldsymbol{H}_1^{(l-1)}, \boldsymbol{A}_1), \text{MSG}_2(\boldsymbol{H}_1^{(l-1)}, \boldsymbol{H}_2^{(l-1)})), \tag{6}$$

where $\text{MSG}_1(\cdot)$ is a message propagation function of vanilla GNNs in Eq. (1). $\text{MSG}_2(\boldsymbol{H}_1^{(l-1)}, \boldsymbol{H}_2^{(l-1)}) = [\mu_1^{(l-1)}, \cdots, \mu_{n_1}^{(l-1)}]^T$ computes cross-graph messages from $\mathcal{G}_2$ to $\mathcal{G}_1$, where each row $\mu_i$ denotes the message from $\mathcal{G}_2$ to the node $i$ in $\mathcal{G}_1$. Let $\boldsymbol{h}_{1,i}^{(l-1)}$ and $\boldsymbol{h}_{2,j}^{(l-1)}$ denote the representation of node $i$ in $\mathcal{G}_1$ and node $j$ in $\mathcal{G}_2$ at layer $l - 1$, respectively. The cross-graph message $\mu_i^{(l-1)}$ is computed by an attention-based module as

$$w_{ji}^{(l-1)} = \frac{\exp(\text{sim}(\boldsymbol{h}_{1,i}^{(l-1)}, \boldsymbol{h}_{2,j}^{(l-1)}))}{\sum_{k=1}^{n_2} \exp(\text{sim}(\boldsymbol{h}_{1,i}^{(l-1)}, \boldsymbol{h}_{2,k}^{(l-1)}))}, \quad \mu_i^{(l-1)} = \sum_{j=1}^{n_2} w_{ji}^{(l-1)}(\boldsymbol{h}_{2,j}^{(l-1)} - \boldsymbol{h}_{1,i}^{(l-1)}), \tag{7}$$

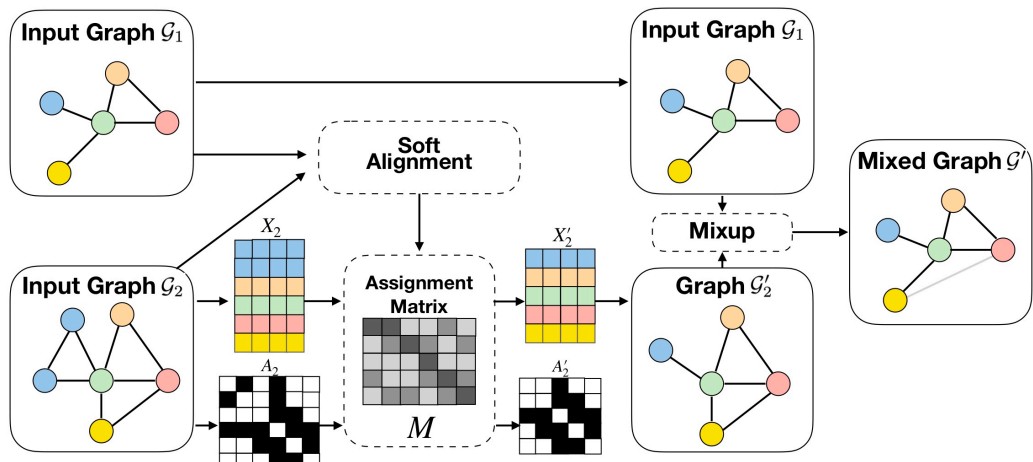

Figure 1: An overview of S-Mixup. Given a pair of graphs $\mathcal{G}_1$ and $\mathcal{G}_2$, the color of each node represents the node-level correspondence between two graphs. The assignment matrix $\boldsymbol{M}$ is obtained by a soft alignment. Based on the assignment matrix, the graph $\mathcal{G}_2'$ is transformed from $\mathcal{G}_2$ to be aligned with $\mathcal{G}_1$. Finally, we obtain the mixed graph $\mathcal{G}'$ by mixing $\mathcal{G}_1$ and $\mathcal{G}_2'$. The darkness level of each edge represents its weight.

where $\text{sim}(\cdot)$ denotes the similarity between two node representations, such as Euclidean distance or cosine similarity. The final node representations $\boldsymbol{H}_1 = \boldsymbol{H}_1^{(L)}$ and $\boldsymbol{H}_2 = \boldsymbol{H}_2^{(L)}$ are obtained after applying $L$ layers of such message passing operations. Given the node representations $\boldsymbol{H}_1$ and $\boldsymbol{H}_2$, we take the likelihood of soft alignment to be proportional to the similarity between node representations. Formally, the soft assignment matrix $\boldsymbol{M}$ is computed as

$$\boldsymbol{M} = \text{softmax}(\text{sim}(\boldsymbol{H}_1, \boldsymbol{H}_2)), \tag{8}$$

where softmax function is a column-wise operation, and $\text{sim}(\cdot)$ computes a similarity score for node pairs between $\mathcal{G}_1$ and $\mathcal{G}_2$.

The network is trained by a triplet loss following Li et al. (2019). Specifically, we treat graphs with the same class label as positive pairs and graphs with different class labels as negative pairs. Intuitively, the learned representations of graphs from the same class should be more similar than those from different classes. To be more specific, at each training step, we first sample a tuple of graphs $(\mathcal{G}_1, \mathcal{G}_2, \mathcal{G}_3)$ from the training dataset. $\mathcal{G}_1$ and $\mathcal{G}_2$ are sampled from the same class, while $\mathcal{G}_3$ is sampled from another class. We use the graph matching network to extract node representations $(\boldsymbol{H}_1, \boldsymbol{H}_2)$ from the graph pair $(\mathcal{G}_1, \mathcal{G}_2)$ as in Eq. (6). Afterwards, the graph representations $\boldsymbol{h}_{\mathcal{G}_1}$ and $\boldsymbol{h}_{\mathcal{G}_2}$ are separately computed from $\boldsymbol{H}_1$ and $\boldsymbol{H}_2$ as in Eq. (2). Similarly, we compute the graph representations $(\boldsymbol{h}_{\mathcal{G}_1}', \boldsymbol{h}_{\mathcal{G}_3})$ of graph pair $(\mathcal{G}_1, \mathcal{G}_3)$. The graph matching network is optimized by minimizing the triplet loss as

$$L_{\text{triplet}} = \mathbb{E}_{(\mathcal{G}_1, \mathcal{G}_2, \mathcal{G}_3)} \max(0, \text{sim}(\boldsymbol{h}_{\mathcal{G}_1}', \boldsymbol{h}_{\mathcal{G}_3}) - \text{sim}(\boldsymbol{h}_{\mathcal{G}_1}, \boldsymbol{h}_{\mathcal{G}_2}) + \gamma), \tag{9}$$

where $\text{sim}(\cdot)$ computes a similarity score between two graph representations. Minimizing such triplet loss encourages the similarity between $\mathcal{G}_1$ and $\mathcal{G}_3$ to be smaller than the similarity between $\mathcal{G}_1$ and $\mathcal{G}_2$ by at least a margin $\gamma$. The graph matching network is first trained on the training data. During the process of mixing graphs, the trained graph matching network is used to compute soft assignment matrices. See Figure 1 for an overview of our proposed S-Mixup method. The implementation details are summarized in Appendix B. There are other well-studied graph alignment methods (Heimann et al., 2018; Zhang & Tong, 2016; Xu et al., 2019; Gold & Rangarajan, 1996) to align the nodes across two graphs. Nevertheless, not all of them are appropriate for our framework due to computational complexity. See more discussion about graph alignment methods in Appendix D.1.

### 3.3 COMPLEXITY ANALYSIS

Given a pair of graphs $\mathcal{G}_1$ with $n_1$ nodes and $\mathcal{G}_2$ with $n_2$ nodes, S-Mixup computes the soft assignment matrix $\boldsymbol{M} \in \mathbb{R}^{n_1 \times n_2}$, thus having a space complexity of $O(n_1 n_2)$. Besides, during the

Table 1: Comparison between ours and other graph mixup methods

| Methods | Instance-level | Preserving motif | Mixing node feature space | Input-level | Perserving graph size |
|---|---|---|---|---|---|
| G-mixup (Han et al., 2022) | | ✓ | | ✓ | |
| Graph Transplant (Park et al., 2022) | ✓ | ✓ | | ✓ | ✓ |
| SubMix (Yoo et al., 2022) | ✓ | | | ✓ | ✓ |
| ifMixup (Guo & Mao, 2021) | ✓ | | ✓ | ✓ | |
| Manifold Mixup (Wang et al., 2021b) | ✓ | | ✓ | | ✓ |
| S-Mixup | ✓ | ✓ | ✓ | ✓ | ✓ |

computation of the soft assignment matrix $M$, the graph matching network uses a cross-graph message passing scheme, which needs to compute attention weights (see Equation (7)) for every pair of nodes across two graphs. Thus, S-Mixup has a computational cost of $O(n_1 n_2)$. This time cost is affordable for small graphs but may lead to large computational and memory costs on large graphs. The better performance of S-Mixup comes from the higher computational cost.

## 4 RELATED WORK

Most commonly used graph data augmentation methods (Velickovic et al., 2019; Rong et al., 2019; Feng et al., 2020; You et al., 2020; Zhu et al., 2020) are based on uniformly random modifications of graph elements, such as dropping edges, dropping nodes, or sampling subgraphs. In addition to random modifications, recent studies (You et al., 2021; Sun et al., 2021; Luo et al., 2022) use a learnable neural network model to automate the selection of augmentation. Another line of research (Suresh et al., 2021; Zhao et al., 2021; Chen et al., 2020; Jin et al., 2020) for improving random modifications is to enhance task-relevant information in augmented graphs with learnable data augmentation methods. However, the above methods are based on a single graph when performing augmentation, so they don't exchange information between different instances. To address the limitation, a few studies propose interpolation-based Mixup methods for graph augmentation. Wang et al. (2021b) follows manifold Mixup (Verma et al., 2019) to interpolate the latent representations of pairs of graphs. Since the graph representations are obtained at the last layer of GNN models, this solution may be not optimal. In contrast to the previous method, ifMixup (Guo & Mao, 2021) interpolates the input graph data instead of latent space. It uses an arbitrary node order to align two graphs and linearly interpolates adjacency matrices and feature matrices to generate new graph data. ifMixup doesn't consider the node-level correspondence between graphs, leading to generating noisy graph data as discussed in Section 5.1. Moreover, the size of the generated graphs equals the larger one of the input pair, resulting in a distribution shift in graph sizes. Unlike ifMixup, Graph Transplant (Park et al., 2022) proposes to generate new graph data by connecting subgraphs from different input graphs. Graph Transplant uses node saliency information to select meaningful subgraphs from input graphs and determine labels of generated graphs. Similarly, SubMix (Yoo et al., 2022) mixes random subgraphs of different input graphs. Nonetheless, random sampling doesn't preserve motifs in the graphs, thus generated graph data may be noisy. Both Graph Transplant (Park et al., 2022) and SubMix (Yoo et al., 2022) only consider graph topology, so the node features of generated graphs are kept the same. Instead of directly mixing instances, G-mixup (Han et al., 2022) proposes a class-level graph mixup method that interpolates the graph generators of different classes. Specifically, it uses graphons to model graph topology structure and then generates synthetic graphs through sampling the mixed graphons of different classes. Note that G-mixup relies on a strong assumption that graphs from the same class can be generated by the same graph generator (i.e., graphon).

However, none of these methods explicitly consider the node-level correspondence between graphs, which is important to generate high-quality graphs as discussed in Section 5.1. In contrast, our approach uses soft graph alignment to compute the node-level correspondence and mixes graphs based on the alignment, thereby avoiding the generation of noisy data. We compare our method with existing graph mixup methods in Table 1.

## 5 DISCUSSION

### 5.1 NODE-LEVEL CORRESPONDENCE MATTERS IN GRAPH MIXUP

We use the MOTIF (Wu et al., 2022) [1] dataset as an example to show the importance of node order. Each graph in the MOTIF dataset is composed of one base (tree, ladder, wheel) and one motif (cycle,

---

[1]MOTIF dataset is a synthetic dataset that is proposed for the out-of-distribution problem. In this work, we avoid introducing the distribution shift when constructing the dataset.

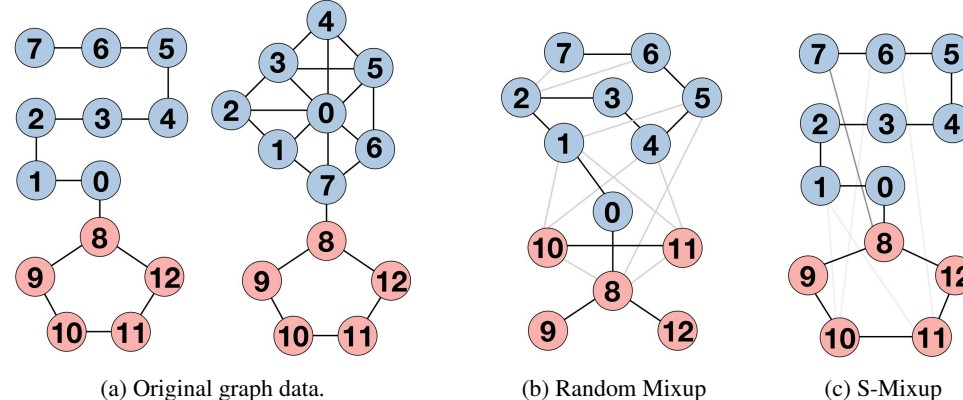

(a) Original graph data.        (b) Random Mixup        (c) S-Mixup

Figure 2: An example from the MOTIF dataset showing that using random order to mix graphs creates noisy data. Red nodes represent the motif, while blue nodes represent the base. The color of the edge indicates its weight. In (a), we show a pair of graphs sampled from the Motif dataset. In both graphs, red nodes represent a cycle motif. In (b), we show the mixed graph using a random node order. In this graph, red nodes no longer form a cycle motif. In (c), we show the mixed graph by our method. In this graph, red nodes still form a cycle motif.

house, crane). The task of the MOTIF dataset is to classify graphs by the motif contained in the graph. In Figure 2, we visualize a case of mixing two graphs in the same class. We randomly align two graphs and linearly interpolate their node feature matrices and adjacency matrices to generate a new graph. As shown in Figure 2b, the generated graph doesn't preserve the motif in the input graphs. In other words, red nodes that form a cycle motif in the original graphs no longer form a cycle motif in the generated graph. Training with such noisy data greatly decreases the accuracy of a GIN (Xu et al., 2018) model from $91.47\%$ to $52.88\%$. The significant performance drop clearly demonstrates the importance of node-level correspondence between graphs when mixing graphs.

## 5.2 GRAPH TRANSFORMATION ANALYSIS

There is a limitation of our method caused by transforming graphs to have the same number of nodes and the same node order. We first introduce graph edit distance (GED) to characterize the similarity of two graphs. Given a pair of graphs $(\mathcal{G}_1, \mathcal{G}_2)$, the graph edit distance $GED(\mathcal{G}_1, \mathcal{G}_2)$ is defined as the minimum cost of an edit path between two graphs. An edit path between graphs $\mathcal{G}_1$ and $\mathcal{G}_2$ is a sequence of edit operations that transforms $\mathcal{G}_1$ to $\mathcal{G}_2$. For graph edit operations, we consider six edit operations, including node insertion, node deletion, node substitution, edge insertion, edge deletion, and edge substitution. The cost of all graph edit operations are given as follows:

- The cost of node insertion and node deletion is defined as the square of the $l_2$ norm of node feature of the inserted node and deleted node, respectively.
- Node substitution is to change the feature of a node and its cost is defined as $||\boldsymbol{x} - \boldsymbol{x}'||_2^2$, where $\boldsymbol{x}$ and $\boldsymbol{x}'$ are the node features before and after node substitution, respectively.
- The cost of edge insertion and edge deletion is the weight of the inserted edge and deleted edge, respectively.
- Edge substitution is to change the weight of an edge and its cost is defined as $|e - e'|$, where $e$ and $e'$ are the weights of the edge before and after edge substitution, respectively.
- The cost of an edit path is defined as the sum of the costs of its operations.

**Definition 1** (Graph Edit Distance (GED)). *For graph pair $(\mathcal{G}_1, \mathcal{G}_2)$, the graph edit distance $GED(\mathcal{G}_1, \mathcal{G}_2)$ is defined as the minimum cost of an edit path between two graphs, i.e., $GED(\mathcal{G}_1, \mathcal{G}_2) = \min_{(op_1, \cdots, op_k) \in \mathbb{P}(\mathcal{G}_1, \mathcal{G}_2)} \sum_{i=1}^{k} c(op_i)$, where $\mathbb{P}(\mathcal{G}_1, \mathcal{G}_2)$ denotes the set of edit paths from $\mathcal{G}_1$ to $\mathcal{G}_2$, $c(op)$ denotes the cost edit operation op.*

Subsequently, we define normalized GED as $\epsilon = \frac{\text{GED}(\mathcal{G}', \mathcal{G}_2)}{\text{GED}(\mathcal{G}', \mathcal{G}_1) + \text{GED}(\mathcal{G}', \mathcal{G}_2)} \in [0, 1]$ to characterize the similarity between the generated graph $\mathcal{G}'$ and the original pair $(\mathcal{G}_1, \mathcal{G}_2)$. Note that, for a perfect mixed result, the normalized GED should be equal to the mixup ratio $\lambda$. To study the difference between normalized GED and mixup ratio, we propose the following theorem.

Table 2: Comparisons between our method and baselines on six datasets from the TUDatasets benchmark with the GIN and GCN model. The average testing accuracy of 10 runs is reported. The best results are shown in bold.

| | Dataset | IMDB-B | PROTEINS | NCI1 | REDDIT-B | IMDB-M | REDDIT-M5 |
|---|---|---|---|---|---|---|---|
| | #graphs | 1000 | 1113 | 4110 | 2000 | 1500 | 4999 |
| | #classes | 2 | 2 | 2 | 2 | 3 | 5 |
| | #avg nodes | 19.77 | 39.06 | 29.87 | 429.63 | 13.00 | 508.52 |
| | #avg edges | 96.53 | 72.82 | 32.30 | 497.75 | 65.94 | 594.87 |
| GCN | Vanilla | $72.80 \pm 4.08$ | $71.43 \pm 2.60$ | $72.38 \pm 1.45$ | $84.85 \pm 2.42$ | $49.47 \pm 2.60$ | $49.99 \pm 1.37$ |
| | DropEdge | $73.20 \pm 5.62$ | $71.61 \pm 4.28$ | $68.32 \pm 1.60$ | $85.15 \pm 2.81$ | $49.00 \pm 2.94$ | $51.19 \pm 1.74$ |
| | DropNode | $73.80 \pm 5.71$ | $72.69 \pm 3.55$ | $70.73 \pm 2.02$ | $83.65 \pm 3.63$ | $50.00 \pm 4.85$ | $47.71 \pm 1.75$ |
| | Subgraph | $70.90 \pm 5.07$ | $67.93 \pm 3.24$ | $65.05 \pm 4.36$ | $68.41 \pm 2.57$ | $49.80 \pm 3.43$ | $47.31 \pm 5.23$ |
| | M-Mixup | $72.00 \pm 5.66$ | $71.16 \pm 2.87$ | $71.58 \pm 1.79$ | $87.05 \pm 2.47$ | $49.73 \pm 2.67$ | $51.49 \pm 2.00$ |
| | SubMix | $72.30 \pm 4.75$ | $72.42 \pm 2.43$ | $71.65 \pm 1.58$ | $85.15 \pm 2.37$ | $49.73 \pm 2.88$ | $52.87 \pm 2.19$ |
| | G-Mixup | $73.20 \pm 5.60$ | $70.18 \pm 2.44$ | $70.75 \pm 1.72$ | $86.85 \pm 2.30$ | $50.33 \pm 3.67$ | $51.77 \pm 1.42$ |
| | S-Mixup | $\mathbf{74.40 \pm 5.44}$ | $\mathbf{73.05 \pm 2.81}$ | $\mathbf{75.47 \pm 1.49}$ | $\mathbf{89.30 \pm 2.69}$ | $\mathbf{50.73 \pm 3.66}$ | $\mathbf{53.29 \pm 1.97}$ |
| GIN | Vanilla | $71.30 \pm 4.36$ | $68.28 \pm 2.47$ | $79.08 \pm 2.12$ | $89.15 \pm 2.47$ | $48.80 \pm 2.54$ | $53.17 \pm 2.26$ |
| | DropEdge | $70.50 \pm 3.80$ | $68.01 \pm 3.22$ | $76.47 \pm 2.34$ | $87.45 \pm 3.91$ | $48.73 \pm 4.08$ | $54.11 \pm 1.94$ |
| | DropNode | $72.00 \pm 6.97$ | $\mathbf{69.64 \pm 2.98}$ | $74.60 \pm 2.12$ | $88.60 \pm 2.52$ | $45.67 \pm 2.59$ | $53.97 \pm 2.11$ |
| | Subgraph | $70.40 \pm 4.98$ | $66.67 \pm 3.10$ | $60.17 \pm 2.33$ | $76.80 \pm 3.87$ | $43.74 \pm 5.74$ | $50.09 \pm 4.94$ |
| | M-Mixup | $72.00 \pm 5.14$ | $68.65 \pm 3.76$ | $79.85 \pm 1.88$ | $87.70 \pm 2.50$ | $48.67 \pm 5.32$ | $52.85 \pm 1.03$ |
| | SubMix | $71.70 \pm 6.20$ | $69.54 \pm 3.15$ | $79.78 \pm 1.09$ | $90.45 \pm 1.93$ | $49.80 \pm 4.01$ | $54.27 \pm 2.92$ |
| | G-Mixup | $72.40 \pm 5.64$ | $64.69 \pm 3.60$ | $78.20 \pm 1.58$ | $90.20 \pm 2.84$ | $49.93 \pm 2.82$ | $54.33 \pm 1.99$ |
| | S-Mixup | $\mathbf{73.40 \pm 6.26}$ | $69.37 \pm 2.86$ | $\mathbf{80.02 \pm 2.45}$ | $\mathbf{90.55 \pm 2.11}$ | $\mathbf{50.13 \pm 4.34}$ | $\mathbf{55.19 \pm 1.99}$ |

**Theorem 1.** *Given a pair of input graphs $\mathcal{G}_1$ and $\mathcal{G}_2$, the mixup ratio is $\lambda$ and the mixed graph is $\mathcal{G}'$. Let $\mathcal{G}_2'$ be the graph transformed based on soft alignments as discussed in Section 3.1. The difference between normalized GED $\epsilon$ and mixup ratio $\lambda$ is upper bounded by*

$$|\epsilon - \lambda| \leq \frac{(1 - \lambda)GED(\mathcal{G}_2, \mathcal{G}_2')}{GED(\mathcal{G}_1, \mathcal{G}_2') + GED(\mathcal{G}_2, \mathcal{G}_2')} \tag{10}$$

Detailed proof of this theorem is given in Appendix A. Note that the difference between normalized GED $\epsilon$ and mixup ratio $\lambda$ equals to zero when input graphs are already aligned (i.e., $\mathcal{G}_2 = \mathcal{G}_2'$). Theorem 1 indicates that such difference is caused by transforming $\mathcal{G}_2$ to $\mathcal{G}_2'$. Furthermore, the difference is small when $\lambda$ is close to 1. Thereby, in this work, we make the range of $\lambda$ to be $[0.5, 1]$ to reduce the difference via taking the maximum value of $\lambda'$ and $1 - \lambda'$. A promising solution is to use an adaptive $\lambda$ range for different graph pairs, and we leave it for future work. See a case study in Appendix F.

## 6 EXPERIMENTS

In this section, we evaluate the effectiveness of our method on six real-world datasets from the TUDatasets benchmark [2] (Morris et al., 2020), including one bioinformatics dataset PROTEINS, one molecule dataset NCI1, and four social network datasets IMDB-BINARY, IMDB-MULTI, REDDIT-BINARY, and REDDIT-MULTI-5K. We first show that in various graph classification tasks, our method substantially improves the performance of different GNN backbones as well as generalization in Section 6.1. Further, we show our method improves the robustness of GNNs against label corruption in Section 6.2. In addition, we conduct extensive ablation studies to evaluate the effectiveness of our design in Section 6.3 and perform a case study in F.

**Baselines.** We compare our methods with the following baseline methods, including (1) DropEdge (Rong et al., 2019), which uniformly removes a certain ratio of edges from the input graphs; (2) DropNode (Feng et al., 2020; You et al., 2020), which uniformly drops a certain portion of nodes from the input graphs; (3) Subgraph (You et al., 2020), which extract subgraphs from the input graphs via a random walk sampler; (4) M-Mixup [3](Verma et al., 2019; Wang et al., 2021b), which linearly interpolates the graph-level representations; (5) SubMix (Yoo et al., 2022), which mixes random subgraphs of input graph pairs; (6) G-Mixup (Han et al., 2022), which is a class-level graph mixup method by interpolating graphons of different classes. For a fair comparison, we use the same architecture of GNNs (e.g., number of layers) and the same training hyperparameters (e.g., learning rate) for all methods. The optimal hyperparameters of all methods are obtained by grid search.

---

[2]https://chrsmrrs.github.io/datasets/docs/datasets/

[3]Although Wang et al. (2021b) proposes mixup methods for both graph and node classification tasks, we only consider the one for graph classification tasks in this paper.

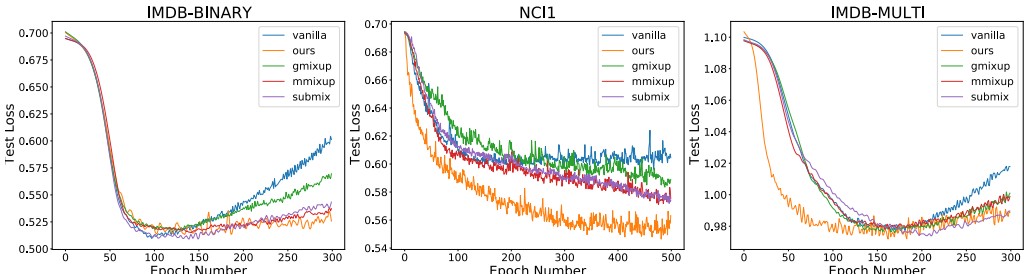

Figure 3: The Learning curves of GCN model on IMDB-BINARY, NCI1, and IMDB-MULTI datasets. The curves are depicted on the averaged test loss over 10 runs.

Table 3: Robustness to label corruption with different ratios.

| Dataset | Methods | 20% | 40% | 60% |
|---------|---------|-----|-----|-----|
| | Vanilla | $69.50 \pm 7.83$ | $62.70 \pm 7.93$ | $45.80 \pm 6.63$ |
| | M-mixup | $70.60 \pm 3.69$ | $64.90 \pm 6.20$ | $47.60 \pm 6.79$ |
| IMDB-B | SubMix | $\mathbf{71.00 \pm 5.23}$ | $62.80 \pm 5.74$ | $48.40 \pm 6.83$ |
| | G-mixup | $69.90 \pm 5.01$ | $63.20 \pm 6.01$ | $48.70 \pm 5.28$ |
| | S-Mixup | $70.20 \pm 5.69$ | $\mathbf{65.10 \pm 5.58}$ | $\mathbf{48.90 \pm 4.61}$ |
| | Vanilla | $48.00 \pm 3.37$ | $44.87 \pm 2.91$ | $36.20 \pm 3.78$ |
| | M-mixup | $48.40 \pm 2.83$ | $44.07 \pm 2.18$ | $38.60 \pm 3.97$ |
| IMDB-M | SubMix | $47.80 \pm 5.16$ | $44.20 \pm 6.75$ | $36.80 \pm 5.44$ |
| | G-mixup | $48.53 \pm 3.08$ | $44.67 \pm 2.42$ | $39.27 \pm 5.12$ |
| | S-Mixup | $\mathbf{49.40 \pm 3.06}$ | $\mathbf{46.27 \pm 3.86}$ | $\mathbf{39.27 \pm 4.57}$ |

**Setup.** We first train the graph matching network until it converges. Then, we evaluate the performance of our method and other baselines by the testing accuracies of a classification model over six datasets. For the classification model, we use two GNN models; namely, GCN (Kipf & Welling, 2016) and GIN (Xu et al., 2018). We split the dataset into train/validation/test data by $80\%/10\%/10\%$. The averaged testing accuracy over 10 runs is reported for comparison. See more experimental details in Appendix C.

### 6.1 S-MIXUP IMPROVES THE PERFORMANCE AND GENERALIZATION

Table 2 summarizes the performance of our proposed S-Mixup compared to baselines on all six datasets. From the results, our method can improve the performance of different GNN models on various datasets. For example, compared to the GCN model without data augmentation, our method achieves an improvement of $4.45\%$, $3.3\%$ and $3.09\%$ on the REDDIT-BINARY, REDDIT-MULTI-5K, and NCI1 datasets, respectively. It is worth noting that our method achieves the best performance among the graph mixup methods. Since M-Mixup only interpolates the graph representations at the last layer of GNN models, its improvement is limited. Meanwhile, G-Mixup generates the same node features for all augmented graphs, thus leading to performance degradation on PROTEINS and NCI1 datasets which have node features.

We further use learning curves to study the effects of mixup methods on GNN models. Figure 3 shows the test loss at each training epoch of our method compared to other graph mixup methods on IMDB-BINARY, NCI1, and IMDB-MULTI datasets. From the results, we have the following observations:

- For the IMDB-BINARY and IMDB-MULTI datasets, the test loss of GCNs without data augmentation increases after certain training epochs. While all graph mixup methods reduce the increase in test loss at later iterations, our method has the best results in helping the GCN model to converge to a lower test loss. This demonstrates that our method can effectively regularize GNN models to prevent over-fitting.

- For the NCI1 dataset, our method achieves an obvious improvement over the other methods. Such observation indicates that our method generates better synthetic graph data than other methods, thereby obtaining a much lower test loss.

Table 4: Results of the ablation study on mixup strategies and $\lambda$ range.

| | IMDB-B | PROTEINS | REDDIT-B | IMDB-M |
|---|---|---|---|---|
| Vanilla | $72.80 \pm 4.08$ | $71.43 \pm 2.60$ | $84.85 \pm 2.42$ | $49.47 \pm 2.60$ |
| S-Mixup w/o different classes | $73.07 \pm 5.78$ | $72.23 \pm 3.56$ | $87.60 \pm 1.74$ | $50.27 \pm 2.86$ |
| S-Mixup with $\lambda \in [0, 1]$ | $73.00 \pm 5.87$ | $72.15 \pm 3.86$ | $88.70 \pm 2.18$ | $49.07 \pm 4.06$ |
| S-Mixup | $\mathbf{74.40 \pm 5.44}$ | $\mathbf{73.05 \pm 2.81}$ | $\mathbf{89.30 \pm 2.69}$ | $\mathbf{50.73 \pm 3.66}$ |

Table 5: Results of the ablation study on normalization.

| | | IMDB-B | PROTEINS | NCI1 | REDDIT-B | IMDB-M | REDDIT-M5 |
|---|---|---|---|---|---|---|---|
| GCN | Sinkhorn | $73.20 \pm 6.13$ | $72.80 \pm 3.72$ | $74.89 \pm 1.34$ | $\mathbf{90.50 \pm 1.80}$ | $50.40 \pm 2.83$ | $53.13 \pm 2.33$ |
| | Softmax | $\mathbf{74.40 \pm 5.44}$ | $\mathbf{73.05 \pm 2.81}$ | $\mathbf{75.47 \pm 1.49}$ | $89.30 \pm 2.69$ | $\mathbf{50.73 \pm 3.66}$ | $\mathbf{53.29 \pm 1.97}$ |
| GIN | Sinkhorn | $73.10 \pm 6.62$ | $69.09 \pm 2.84$ | $79.95 \pm 2.15$ | $\mathbf{91.25 \pm 1.70}$ | $\mathbf{50.23 \pm 4.70}$ | $55.09 \pm 1.54$ |
| | Softmax | $\mathbf{73.40 \pm 6.26}$ | $\mathbf{69.37 \pm 2.86}$ | $\mathbf{80.02 \pm 2.45}$ | $90.55 \pm 2.11$ | $50.13 \pm 4.34$ | $\mathbf{55.19 \pm 1.99}$ |

## 6.2 S-MIXUP IMPROVES THE ROBUSTNESS

In this subsection, we evaluate the robustness of our method and other graph mixup methods against noisy labels. We generate the noisy training data by randomly corrupting the labels of the IMDB-BINARY and IMDB-MULTI datasets. Specifically, we create three training datasets, where $20\%$, $40\%$, and $60\%$ of the labels are flipped to a different class, respectively. All the test labels are kept the same for evaluation. We adopt GCN as the classification backbone in this experiment. Results in Table 3 show that our method can achieve the best performance in most cases, indicating that our method can improve the robustness of GNNs against corrupted labels.

## 6.3 ANALYSIS

**Mixup strategy.** In this subsection, we investigate the performance of different mixup strategies. Specifically, we use "S-Mixup w/o different classes" to denote a design choice that only mixes graphs from the same class. In this experiment, we adopt GCN as the classification model on IMDB-BINARY, PROTEINS, REDDIT-BINARY, and IMDB-MULTI datasets. Results in Table 4 demonstrate that "S-Mixup w/o different classes" creates more training graph data, leading to the improvement in the performance of GNN models. In addition, mixing graphs from different classes can further improve the performance of GNN models. Such observation shows that mixing all the classes is a better design choice for the mixup strategies.

**Mixup ratio $\lambda$.** As we mentioned in Section 5.2, the $\lambda$ range has an effect on the quality of the generated graphs, so we investigate the performance of different $\lambda$ ranges in this subsection. Specifically, we compare the performance of $\lambda \in [0, 1]$ and the default setting (i.e., $\lambda \in [0.5, 1]$). In this experiment, we adopt GCN as the classification model on IMDB-BINARY, PROTEINS, REDDIT-BINARY, and IMDB-MULTI datasets. Table 4 shows that our method achieves better performance with larger $\lambda$ value, which is consistent with our analysis in Section 5.2.

**Softmax normalization.** While most graph matching algorithms use sinkhorn normalization to fulfills the requirements of doubly-stochastic alignment, we relax this constraint by only applying column-wise softmax normalization on the soft assignment matrix $M$ in Equation (8). In this subsection, we investigate the effect of different normalization functions on our framework. Specifically, we replace softmax normalization in Equation (8) with sinkhorn normalization. Results in Table 5 show that sinkhorn normalization has a similar performance to softmax normalization. Since sinkhorn normalization has a higher computational cost than softmax normalization, we choose to use softmax normalization in our framework for efficiency.

## 7 CONCLUSION

In this work, we propose S-Mixup, a novel mixup method for graph classification by soft graph alignments. S-Mixup computes a soft assignment matrix to model the node-level correspondence between graphs. Based on the soft assignment matrix, we transform one graph to align with the other graph and then interpolate adjacency matrices and node feature matrices to generate augmented training graph data. Experimental results demonstrate that our method can improve the performance and generalization of GNNs, as well as the robustness of GNNs against noisy labels. In the future, we would like to apply S-Mixup to other tasks on graphs, such as the node classification problem.

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

## A    PROOF OF THEOREM 1

*Proof.* Since $\mathcal{G}'$ has the same number of nodes and the same node order as $\mathcal{G}_1$, we have

$$
\begin{aligned}
\text{GED}(\mathcal{G}', \mathcal{G}_1) &= \phi(\boldsymbol{A}' - \boldsymbol{A}_1) + ||\boldsymbol{X}' - \boldsymbol{X}_1||_F^2 \\
&= (1 - \lambda)\phi(\boldsymbol{A}_2' - \boldsymbol{A}_1) + (1 - \lambda)||\boldsymbol{X}_2' - \boldsymbol{X}_1||_F^2 \\
&= (1 - \lambda)\text{GED}(\mathcal{G}_1, \mathcal{G}_2')
\end{aligned}
\tag{11}
$$

where $\phi(\boldsymbol{A})$ calculates the sum of the absolute value of all elements in the matrix $\boldsymbol{A}$. Similarly, we have

$$
\text{GED}(\mathcal{G}', \mathcal{G}_2') = \lambda\text{GED}(\mathcal{G}_1, \mathcal{G}_2')
\tag{12}
$$

From the triangle inequality, we have

$$
\text{GED}(\mathcal{G}', \mathcal{G}_2) \leq \text{GED}(\mathcal{G}', \mathcal{G}_2') + \text{GED}(\mathcal{G}_2, \mathcal{G}_2')
\tag{13}
$$

Then the difference between normalized GED and mixup ratio is

$$
\begin{aligned}
|\epsilon - \lambda| &= \frac{\text{GED}(\mathcal{G}', \mathcal{G}_2)}{\text{GED}(\mathcal{G}', \mathcal{G}_1) + \text{GED}(\mathcal{G}', \mathcal{G}_2)} - \lambda \\
&\overset{(a)}{\leq} \frac{\text{GED}(\mathcal{G}', \mathcal{G}_2') + \text{GED}(\mathcal{G}_2, \mathcal{G}_2')}{\text{GED}(\mathcal{G}', \mathcal{G}_1) + \text{GED}(\mathcal{G}', \mathcal{G}_2') + \text{GED}(\mathcal{G}_2, \mathcal{G}_2')} - \lambda \\
&\overset{(b)}{=} \frac{\text{GED}(\mathcal{G}', \mathcal{G}_2') + \text{GED}(\mathcal{G}_2, \mathcal{G}_2')}{(1 - \lambda)\text{GED}(\mathcal{G}_1, \mathcal{G}_2') + \text{GED}(\mathcal{G}', \mathcal{G}_2') + \text{GED}(\mathcal{G}_2, \mathcal{G}_2')} - \lambda \\
&\overset{(c)}{=} \frac{\lambda\text{GED}(\mathcal{G}_1, \mathcal{G}_2') + \text{GED}(\mathcal{G}_2, \mathcal{G}_2')}{(1 - \lambda)\text{GED}(\mathcal{G}_1, \mathcal{G}_2') + \lambda\text{GED}(\mathcal{G}_1, \mathcal{G}_2') + \text{GED}(\mathcal{G}_2, \mathcal{G}_2')} - \lambda \\
&= \frac{\lambda\text{GED}(\mathcal{G}_1, \mathcal{G}_2') + \text{GED}(\mathcal{G}_2, \mathcal{G}_2')}{\text{GED}(\mathcal{G}_1, \mathcal{G}_2') + \text{GED}(\mathcal{G}_2, \mathcal{G}_2')} - \lambda \\
&= \frac{(1 - \lambda)\text{GED}(\mathcal{G}_2, \mathcal{G}_2')}{\text{GED}(\mathcal{G}_1, \mathcal{G}_2') + \text{GED}(\mathcal{G}_2, \mathcal{G}_2')}
\end{aligned}
\tag{14}
$$

where inequality (a) holds due to equation (11), inequality (b) holds due to equation (12), and (c) holds due to equation (13). □

## B    IMPLEMENTATION DETAILS

In this section, we provide the implementation details for our method. We first present the pseudo codes for training the graph matching network in Algorithm 1. After training the graph matching network, we use it to perform Mixup on graphs. The pseudo codes for mixing up graphs are summarized in Algorithm 2. The mixed graphs are used as the augmented training data. To reduce I/O cost, we follow Zhang et al. (2017) to mix graphs from the same batch by random shuffling. For the mixup ratio, we select the hyperparameter $\alpha$ from $\{0.1, 0.2, 0.5, 1, 2, 5, 10\}$. For the sim function, we consider two different metrics, including cosine similarity and Euclidean distance. The optimal hyperparameters are obtained by grid search.

---

**Algorithm 1** Training algorithm

---

**Input:** a training set $\mathbb{S}$, graph matching network GMNET

    **while** not converged **do**

        Sample a tuple of graphs $(\mathcal{G}_1, \mathcal{G}_2, \mathcal{G}_3)$ from $\mathbb{S}$, where $\mathcal{G}_1$ and $\mathcal{G}_2$ are sampled from the same class and $\mathcal{G}_3$ is sampled from another class.

        Obtain a pair of graph representations $(\boldsymbol{h}_{\mathcal{G}_1}, \boldsymbol{h}_{\mathcal{G}_2})$ from the graph pair $(\mathcal{G}_1, \mathcal{G}_2)$ using GMNET

        Obtain a pair of graph representations $(\boldsymbol{h}_{\mathcal{G}_1'}, \boldsymbol{h}_{\mathcal{G}_3})$ from the graph pair $(\mathcal{G}_1, \mathcal{G}_3)$ using GMNET

        Compute $L_{\text{triplet}}$ as Eq. (9)

        Update GMNET by applying stochastic gradient descent to minimize $L_{\text{triplet}}$

    **end while**

---

---

**Algorithm 2** Mixup algorithm

---

**Input:** well-trained graph matching network GMNET, a pair of graphs $(\mathcal{G}_1, \mathcal{G}_2)$
  Generate a pair of node representations $(\boldsymbol{H}_1, \boldsymbol{H}_2)$ from graph pair $(\mathcal{G}_1, \mathcal{G}_2)$ using GMNET
  Compute the soft assignment matrix as Eq. (8)
  Generate the synthetic graph $\mathcal{G}'$ and corresponding label $\boldsymbol{y}'$ as Eq. (5)
  **return** $\mathcal{G}'$ and $\boldsymbol{y}'$

---

| Datasets | Initial learning rate | # Training epochs | Batch size |
|---|---|---|---|
| IMDB-B | 0.001 | 300 | 256 |
| PROTEINS | 0.001 | 300 | 256 |
| NCI1 | 0.01 | 500 | 256 |
| REDDIT-B | 0.01 | 500 | 16 |
| IMDB-M | 0.001 | 300 | 256 |
| REDDIT-M5 | 0.01 | 500 | 16 |

Table 6: Hyperparameters for training classification model

## C EXPERIMENTAL DETAILS

### C.1 EXPERIMENTAL SETTING

For a fair comparison, we use the same architecture of GNN backbones and the same training hyper-parameters for all methods. For the classification model, we use two GNN models; namely, GCN and GIN. The details of GNNs are listed as follows,

- **GCN**(Kipf & Welling, 2016). The number of GCN layers is four, and we use a global mean pooling as the readout function. We set the hidden size as 32. The activation function is ReLU.
- **GIN**(Xu et al., 2018). We use a global mean pooling as the readout function. The number of GIN layers is four, and all MLPs have two layers. We set the hidden size as 32. The activation function is ReLU.

We use the Adam optimizer (Kingma & Ba, 2015) to train all models. See Table 6 for the hyperparameters of training the classification model. We split the dataset into train/validation/test data by $80\%/10\%/10\%$. The best model is selected on the validation set.

For the graph matching network used in S-Mixup, we set the hidden size as 256 and the readout layer as global sum pooling. For all six datasets, the graph matching network is trained for 500 epochs with a learning rate of 0.001. For the number of layers and batch size, see Table 7.

| Datasets | # layers | Batch size |
|---|---|---|
| IMDB-B | 6 | 256 |
| PROTEINS | 5 | 256 |
| NCI1 | 5 | 256 |
| REDDIT-B | 4 | 8 |
| IMDB-M | 5 | 256 |
| REDDIT-M5 | 4 | 8 |

Table 7: Hyperparameters for the graph matching network

## D MORE DISCUSSIONS

### D.1 GRAPH MATCHING METHODS

There are many well-studied graph match methods, which aim to find the node-level correspondence between the graph pairs. Traditional graph matching problem is often formulated as a quadratic assignment problem (QAP). For example, Gold & Rangarajan (1996) relaxes the constraints and proposes the graduate assignment algorithm to solve the QAP. Leordeanu & Hebert (2005) uses spectral

Table 8: The sensitivity of S-Mixup to mixup hyperparameter $\alpha$.

| $\alpha$ | 0.1 | 0.2 | 0.5 | 1 | 2 | 5 | vanilla |
|---|---|---|---|---|---|---|---|
| REDDIT-B | 89.10$\pm$ 2.09 | 89.15 $\pm$ 2.10 | **89.30 $\pm$ 2.69** | 84.70 $\pm$ 6.17 | 81.40 $\pm$ 8.01 | 77.25 $\pm$ 5.39 | 84.85 $\pm$ 2.42 |
| IMDB-M | 50.47 $\pm$ 3.38 | **50.73 $\pm$ 3.66** | 49.67 $\pm$ 2.89 | 50.53$\pm$2.77 | 50.29 $\pm$ 3.91 | 49.20 $\pm$ 3.34 | 49.47 $\pm$ 2.60 |

relaxation to approximate the QAP. The Spectral Matching first relaxes the integer constraints and uses a greedy method to satisfy them later. Furthermore, they propose an iterative matching method IPFP (Leordeanu et al., 2009) to optimize quadratic score in the discrete domain with climbing and convergence properties. Unlike these methods, Cho et al. (2010) proposes a novel random walk algorithm to solve the matching problem. However, it usually takes a long time for these traditional algorithms (Wang et al., 2017; Dokeroglu & Cosar, 2016) to compute the alignment, thereby we don't use traditional graph matching algorithms in our framework for efficiency.

Another line of research uses learning-based networks to compute the alignments between graphs. For example, Li et al. (2019) proposes to extract node embeddings by a new cross-graph attention-based matching mechanism. Xu et al. (2019) learns the alignment and node embeddings of graphs simultaneously with a Gromov-Wasserstein learning framework. Unlike these methods that obtain the final alignment by computing the pairwise similarity between node embeddings, Fey et al. (2020) proposes to add a second stage to iteratively update the initial alignments. Wang et al. (2021a) propose to learn the association graph, so the matching problem is translated into a constrained node classification task. While many learning-based graph matching methods use ground truth correspondences to supervise the training of the networks, in our problem, there is a lack of such ground truth correspondences. Only a few studies address the problem by using self-supervised learning. For example, Liu et al. (2022) proposes a contrastive learning framework for the visual graph matching problem. In our work, we follow Li et al. (2019) to use the triplet loss in Equation 9 to train the graph matching network by incorporating the prior knowledge that the learned representations of graphs from the same class should be more similar than those from different classes.

## D.2 GRAPH EDIT DISTANCE

Sanfeliu & Fu (1983) first introduces the graph edit distance (GED) to measure the similarity between graph pairs. As we introduced in Section 5.2, GED is defined as the minimum cost of an edit path which is a sequence of elementary graph transformation operations. Computing the GED is known as an NP-hard problem (Bunke, 1997) and many methods have been proposed to reduce the computational cost of GED. For example, Riesen & Bunke (2009) approximates the GED computation by means of bipartite graph matching. Several studies (Bougleux et al., 2017; Neuhaus & Bunke, 2007) show that GED is closely related to QAP and can be computed efficiently by graph matching solvers. Besides, there are some learning-based methods (Peng et al., 2021) to improve GED computation. We believe they can be adopted in our framework to compute an adaptive $\lambda$ range for different graph pairs, addressing the limitation of S-Mixup as discussed in Section 5.2.

## D.3 MIXUP FOR SELF-SUPERVISED LEARNING

While we study mixup methods for graph classification problems in this paper, a few studies have investigated mixup in self-supervised learning on graphs. For example, Verma et al. (2021) proposes Mixup-noise to generate positive and negative samples for contrastive learning. Zhang et al. (2022) proposes to generate negative samples for contrastive learning by mixing multiple samples with adaptive weights.

## E SENSITIVITY ANALYSIS TO MIXUP HYPERPARAMETER

In this section, we conduct a hyperparameter study to analyze the sensitivity of S-Mixup to mixup hyperparameter $\alpha$. We sample $\lambda'$ from beta distribution parameterized by different $\alpha$. Specifically, we tune hyperparameter $\alpha$ among {0.1,0.2,0.5,1,2,5}. We adopt GCN as the classification backbone in this experiment. Results in Table 8 show that $\alpha \in [0.1, 0.5]$ consistently leads to better performance than vanilla, while using too large $\alpha$ may lead to underfitting. After we tune the hyperparameter $\alpha$, S-Mixup significantly improves GNNs' performance.

# F    WHAT IS S-MIXUP DOING? A CASE STUDY

In this subsection, we study the outcomes of S-Mixup via a case study on the MOTIF dataset. Firstly, we investigate the case of mixing graphs from the same class. We select a random pair of graphs from the same class in the MOTIF dataset and visualize the outcomes of our method in Figure 2. The original graphs and the generated graph are shown in Figure 2a and Figure 2c, respectively. The results show that our method can preserve the motif in the mixed result. In other words, the red nodes still form a cycle motif in the generated graph.

To study the case of mixing graphs from different classes, we characterize the similarity between generated graphs and original graph pairs using the normalized GED $\epsilon$ introduced in Section 5.2. In this case study, we select a random pair of graphs from different classes in the MOTIF dataset and visualize the relationship between the mixup ratio $\lambda$ and the normalized GED in Figure 4. We observe that the normalized GED is closer to the mixup ratio $\lambda$ when $\lambda$ is larger. Such observation is consistent with Theorem 1 in Section 5.2.

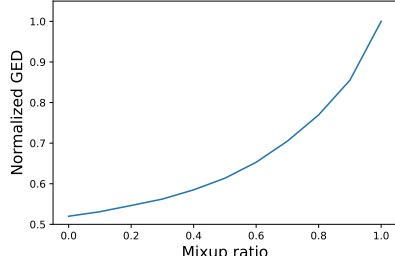

Figure 4: Relationship between normalized GED and mixup ratio $\lambda$

