# OpenReview forum: "Graph Mixup with Soft Alignments"
_ICLR.cc/2023/Conference — Submitted to ICLR 2023_

### Official Review · Reviewer_bSxi · 2022-10-24

**Confidence:** 4
**Correctness:** 3
**Technical Novelty And Significance:** 2
**Empirical Novelty And Significance:** 2
**Recommendation:** 3

**Clarity, Quality, Novelty And Reproducibility:**

The paper is well-written and clear.\
The novelty is limited and since the method is straightforward, I believe reproducibility wouldn't be an issue here.

**Strength And Weaknesses:**

**Strengths**:
- The paper is well-written and clear.
- The method is straightforward and easy to implement.


**Weaknesses**:
- Pairwise node correspondence calculation of the graphs is expensive. There should be a thorough time and computation complexity analysis between the proposed method and the baselines.
- The performance improvement is incremental with respect to the baseline method, especially when the encoder is slightly more powerful (GIN instead of GCN). This raises the question that with large graphs and more powerful graph encoders, how much S-mixup can be practically useful.



**Summary Of The Paper:**

This paper proposes S-mixup, an approach for using the well-known mixup methods on graphs. The authors present calculating an assignment matrix for the corresponding nodes between the two graphs so that they can use the image-like mixup approach in this domain.

**Summary Of The Review:**

Most of the points are mentioned in the strength and weaknesses section but in summary, although S-mixup shows performance improvements by considering the calculation cost and the limited novelty, I think that the method isn't ready for being published at this stage.

---

> ### Author Response · Authors · 2022-11-18
> **Responses to reviewer bSxi**
>
> Thanks for the constructive comments. We have revised the manuscript accordingly and provided pointwise responses below.
>
> >**Q1. There should be a thorough time and computation complexity analysis.**
>
> Thanks for the constructive comment. Given a pair of graphs $G_1$ with $n_1$ nodes and $G_2$ with $n_2$ nodes, S-Mixup computes the soft assignment matrix $M \in R^{n_1 \times n_2}$, thus having a space complexity of $O(n_1 n_2)$. Besides, the graph matching network computes attention weights (see Equation (9)) for every pair of nodes across two graphs. Thus, S-Mixup has a computational cost of $O(n_1n_2)$. We have added a section about complexity analysis in Section 3.3 of the revised manuscript.
>
> The time complexity of baselines is as follows. For M-Mixup, since it only interpolates the graph-level representations, it has the same time complexity. For SubMix, it has a time complexity of $O(E_1 + E_2)$, where $E_1$ and $E_2$ are the number of edges in graphs $G_1$ and $G_2$, respectively. For G-Mixup, it computes graphon and samples new graphs during preprocessing. Thus, it doesn't need extra time when training the classifier model. Compare to these baselines, the better performance of the S-Mixup comes from the higher computational cost.
>
>
> >**Q2. The performance is incremental, especially when the encoder is slightly more powerful (GIN instead of GCN).**
>
> We respectfully disagree with this.
>
> - First, Table 2 shows that compared with the state-of-the-art graph mixup methods, our S-Mixup consistently outperforms them on six benchmarks using two different GNN backbones.
>
> - Besides, on IMDB-BINARY and IMDB-MULTI datasets, S-Mixup improves the performance of GIN more than that of GCN. Thus, we believe that the improvement brought by our S-Mixup is not related to the expressive power of the GNN backbones.
>
> - More importantly, improving the performance of GNNs is only a part of our S-Mixup. As shown in Figure 3, S-Mixup can regularize GNN models to prevent over-fitting. Furthermore, results in Table 3 show that our S-Mixup can improve the robustness of GNNs against corrupted labels better than other graph mixup methods.
>
> >**Q3. The novelty is limited and the method is straightforward.**
>
> We respectfully disagree with this.
>
> - First, although our approach is simple, applying mixup on graphs is not trivial. The key challenge lies in the fact that **different graphs typically have different numbers of nodes**. Even for graphs with the same number of nodes, there **lacks a node-level correspondence that is required to perform mixup.** While previous graph mixup methods use various tricks to sidestep this problem,
> we are the first to explicitly model the node-level correspondences when performing graph mixup.
>
> - In addition, in Section 5.1, we use the MOTIF dataset to **study the importance of the node-level correspondences** between different graph pairs when mixing graphs. In addition to the visualization of the case study, experimental results on the MOTIF dataset also show that **graph mixup with random alignment can hurt motifs and generate noisy data**, leading to a decrease in the performance of the GIN model. To this end, we believe that explicitly considering node-level correspondences between graph pairs in graph mixup is important and convincing.
>
> - Furthermore, we conduct systematic experiments to show our method can consistently outperform the state-of-the-art graph mixup methods.

---

### Official Review · Reviewer_KdGK · 2022-10-25

**Confidence:** 4
**Correctness:** 3
**Technical Novelty And Significance:** 3
**Empirical Novelty And Significance:** 3
**Recommendation:** 6

**Clarity, Quality, Novelty And Reproducibility:**

* Clarity: Overall, the method is simple, and the paper reads well.
* Quality: The paper compares the proposed method with mixup techniques for graphs. The main idea is simple, interesting, and more importantly, effective.
* Novelty: It has some interesting contributions, although the framework is not general enough to apply other tasks on graphs.
* Reproducibility: The pipeline clearly was introduced, and I believe that it is reproducible.

**Strength And Weaknesses:**

Strengths:
* The paper is well-written, and it has a comprehensive discussion about related work and the important aspects of data augmentation for graph-structured datasets.
* This paper theoretically analyzes the discrepancy between the Mixup ratio and graph edit distance. It is not clear whether data augmentation (or Mixup) respecting Graph Edit Distance will be more effective or not. But it is a start to explore Mixup with graph-related distance measures.
* The proposed method is simple and effective, and easy to implement. Compared to strong/recent baselines, the proposed method shows competitive performance.

Weaknesses:
* In data augmentation, controlling the strength and diversity of augmentation is crucial. In this sense, preserving graph size and motif may limit the diversity of augmentation. Graph size preserving is not always desirable. In some applications, one may want to generate samples with diverse graph sizes. This paper needs discussion about it.
* The proposed method needs to be compared with broader related work such Hu et al. [1] GraphMix [2], Gaug [3] etc.
* Figure 2, (c) has links between red and blue nodes. It can be viewed that the cycle motif is broken. Does motif preserving mean here keep all the edges of one of the original graphs? So, unlike standard Mixup, in this framework, the information from other graphs can be added to the target sample as additive noise on the adjacency matrix and features. So, then is no edge/node drop of the target graph possible in this framework?

[1] Hu, Weihua, et al. "Strategies for pre-training graph neural networks." arXiv preprint arXiv:1905.12265 (2019).
[2] Verma, Vikas, et al. "Graphmix: Improved training of gnns for semi-supervised learning." Proceedings of the AAAI Conference on Artificial Intelligence. Vol. 35. No. 11. 2021.
[3] Zhao, Tong, et al. "Data augmentation for graph neural networks." Proceedings of the AAAI Conference on Artificial Intelligence. Vol. 35. No. 12. 2021.

Questions:
1.  In figure 3, the IMDB-BINARY case, vanilla achieved the best performance at the 100th epoch. How was the model selection performed? One purpose of data augmentation is regularization to prevent overfitting, but early stopping or model selection are handy ways to address this problem. If the model at the last epoch was selected for evaluation, it might be misleading.
2. The proposed method is limited to class-level tasks. Is it extensible to other tasks on graphs beyond graph classification?
3. In eq (4), M is used. Did you consider the square root of M as in the normalized graph Laplacian matrix?
4. What alpha was used for the beta distribution? How did you tune the hyperparameter? How sensitive is the performance gain to the hyperparameter tuning?
5. How do you make sure that the augmented samples have the same labels? Or do you interpolate labels as well?


**Summary Of The Paper:**

This paper proposes a new data augmentation method for graph-structured data. Unlike Euclidean spaces, on graphs, it is difficult to define interpolation due to the lack of mapping between two different spaces. Using soft alignments, the authors map one graph to the other and perform Mixup. The authors claim that the proposed method preserves important subgraphs (motifs) and the graph size. Also, unlike manifold Mixup, it performs data augmentation in the input data space at an instance level.

**Summary Of The Review:**

Overall, this paper is well-written and studies the less-explored problem, data augmentation for graphs. The framework is simple and effective. The properties and behaviors of the proposed method have been well analyzed. More qualitative analyses comparing augmented samples by baselines will be useful to understand the value of this work. Also, the limitation of this work is not well discussed like when this work fails and when this work is effective. With some minor updates incorporating reviewers' comments, this paper can be further improved.

---

> ### Author Response · Authors · 2022-11-18
> **Responses to reviewer KdGK part 2/2**
>
>
> >**Q4. How was the model selection performed?**
>
> For every data split, the model with the best performance on the evaluation dataset is selected for testing. The average test results of ten different splits are reported in Table 2. In other words, for the case of IMDB-BINARY, the vanilla model around the 100th epoch is selected for testing.
>
> >**Q5. The proposed method is limited to class-level tasks. Is it extensible to other tasks on graphs beyond graph classification?**
>
> Our method is designed for graph classification tasks. In future work, we will extend S-Mixup to other tasks on graphs, such as the node classification problem.
>
> >**Q6. In eq (4), M is used. Did you consider the square root of M as in the normalized graph Laplacian matrix?**
>
> We don't consider the square root of $M$. Instead, we use the softmax function to perform normalization on $M$.
>
> >**Q7. Hyperparameters of beta distribution.**
>
> As we mentioned in Appendix B, the optimal mixup hyperparameter α is obtained by **grid search** among \{0.1, 0.2, 0.5, 1, 2, 5, 10\}. In addition, we have conducted ablation studies on α and added them to Appendix E of the revised manuscript. The experiment results in the following table (Table 8 in the revised manuscript) show that $\alpha \in [0.1,0.5]$ consistently leads to better performance than vanilla.
>
> |α|0.1|0.2|0.5|1|2|5|vanilla|
> |----|----|----|----|----|----|----|----|
> |REDDIT-B| 89.10$\pm$ 2.09 | 89.15 $\pm$ 2.10 | **89.30 $\pm$ 2.69** | 84.70 $\pm$ 6.17 | 81.40 $\pm$ 8.01 | 77.25 $\pm$ 5.39 |  84.85 $\pm$ 2.42
> |IMDB-M|  50.47 $\pm$ 3.38 | **50.73 $\pm$ 3.66**  | 49.67 $\pm$ 2.89 | 50.53$\pm$2.77 | 50.29 $\pm$ 3.91 | 49.20 $\pm$ 3.34 | 49.47 $\pm$ 2.60
>
> >**Q8. How do you make sure that the augmented samples have the same labels? Or do you interpolate labels as well?**
>
> We don't make the augmented samples have the same labels. Instead, we believe interpolating labels is the key to the success of Mixup. Thus, **as we mentioned in Equation (5), we interpolate the labels as well**.
>
> >Reference
>
> [1] Hu, Weihua, et al. "Strategies for pre-training graph neural networks." arXiv preprint arXiv:1905.12265 (2019).
>
> [2] Verma, Vikas, et al. "Graphmix: Improved training of gnns for semi-supervised learning." Proceedings of the AAAI Conference on Artificial Intelligence. Vol. 35. No. 11. 2021.
>
> [3] Zhao, Tong, et al. "Data augmentation for graph neural networks." Proceedings of the AAAI Conference on Artificial Intelligence. Vol. 35. No. 12. 2021.

---

> ### Author Response · Authors · 2022-11-18
> **Responses to reviewer KdGK part 1/2**
>
> Thanks for the constructive comments. We have revised the manuscript accordingly and provided pointwise responses below.
>
> >**Q1. Preserving graph size and motif may limit the diversity of augmentation.**
>
> Although preserving graph size may limit the diversity of augmentation,
>  **it leads to preserving the training distribution of graph sizes**. We believe it is desirable to keep the distribution of the dataset consistent after data augmentation. Otherwise, if the distribution of augmented data is different from the original data, this can lead to out-of-distribution (OOD) problems, which may degrade the performance of models trained on augmented data.
>
> Besides, when mixing graphs from the same class, **preserving motifs avoids generating noisy data**. When mixing graphs from different classes, the generated graphs may contain motifs from different classes simultaneously, thereby increasing the diversity of augmentation.
>
> >**Q2. The proposed method needs to be compared with broader related work.**
>
> Our proposed S-Mixup method is a graph data augmentation method for graph classification, so we compare our methods with the most commonly used graph augmentation methods and the state-of-the-art graph mixup methods. We believe the problems studied in the related work you mentioned are **different** from ours for the following reasons.
>
> - Hu et al. [1] propose self-supervised methods for pretraining GNNs. We believe pre-training and data augmentation are not mutually exclusive. Instead, they can be combined to improve the performance of neural networks.
>
> - GraphMix [2] proposes a regularization method for GNNs on semi-supervised node classification, while we consider graph classification in this paper. We believe it is out of the scope of this paper to compare with GraphMix.
>
> - Gaug [3] proposes a learnable graph augmentation method for node classification. Since our methods consider graph classification, we believe it is out of the scope of this paper to compare with Gaug.
>
> In future work, we would like to extend S-Mixup to other graph representation tasks, such as the node classification problem. We added a discussion of future work in Section 7 of the revised manuscript.
>
> >**Q3. Question about motif preserving in Figure 2.**
>
> Motif preserving means the subgraph that can determine the class label still exists in the augmented data. For example, in figure 2 (a), the connection of five red nodes forms a cycle structure (8->9->10->11->12->8). In figure 2 (c), the connection of the five red nodes still forms a cycle structure (8->9->10->11->12->8). Although there are several edges connecting blue and red nodes (e.g., 6<->10), these edges have low weights and don't break the cycle motif.

---

### Official Review · Reviewer_pxKa · 2022-11-03

**Confidence:** 5
**Clarity, Quality, Novelty And Reproducibility:** Good.
**Correctness:** 4
**Technical Novelty And Significance:** 3
**Empirical Novelty And Significance:** 2
**Recommendation:** 6

**Strength And Weaknesses:**

Pros:

1) The paper is clearly written and the idea is interesting and novel.
2) The experimental results are convincing with state-of-the-art performance on public benchmarks.
3) The current setting is basically supervised learning, while I believe it can be extended to general cases like self-supervised learning (see the following suggestion on related works ECCV22, SIGKDD22).

Cons:

I did not find any specific concerns about the technical part of the paper. While the paper may acknowledge some more existing works to better position this work.

I have some suggestions on expanding the related work part (if space permits or put some to the appendix).

In the related work part, the authors may also briefly discuss:
1) the works on graph matching, whose purpose is finding node correspondence either softly or in a hard manner. The authors may not be aware a recent ECCV22 paper [a] on graph augmentation for self-supervised graph matching, which I think is worth mentioning as it also augments multiple copies of graph with node correspondence.

[a] Self-supervised Learning of Visual Graph Matching, ECCV 2022

2) As graph edit distance (which is a more flexible technique to expand the mixup space) is also used in the paper, such related works are also needed to discuss in related work part. There are some recent papers on learning-based graph edit distance computing.

3) For mix-up based graph augmentation, there is another missed reference:
[b] m-mix: Generating Hard Negatives via Multi-sample Mixing for Contrastive Learning, SIGKDD 2022

**Summary Of The Paper:**

The paper proposes a node assignmet based technique for graph data mixup, as mixup is a standard, popular and effective way of data augmentation for images, and its effectiveness is also recently explored in the graph area. The experimental results are convincing.

**Summary Of The Review:**

See above comments.

---

> ### Author Response · Authors · 2022-11-18
> **Responses to reviewer pxKa**
>
> Thanks for your review and constructive comments. We have revised the paper accordingly.
>
> >**The paper may acknowledge some more existing works to better position this work.**
>
> We add the discussion about graph matching methods and include the ECCV22 paper [1] in Appendix D.1 of the revised manuscript. Specifically, we discuss both traditional graph matching methods and learning-based graph matching methods and the reason why not all of these well-developed algorithms are appropriate for our framework.
> Besides, We add the discussion about graph edit distance in Appendix D.2 of the revised manuscript. For the KDD paper [2], we add the discussion about mixup for self-supervised learning in Appendix D.3.
>
> >Reference
>
> [1] Liu et al. "Self-supervised Learning of Visual Graph Matching." ECCV, 2022.
>
> [2] Zhang et al. "M-Mix: Generating Hard Negatives via Multi-sample Mixing for Contrastive Learning." SIGKDD, 2022.

---

### Official Review · Reviewer_t5LQ · 2022-11-03

**Confidence:** 4
**Clarity, Quality, Novelty And Reproducibility:** This paper is well-written and easy t…
**Correctness:** 3
**Technical Novelty And Significance:** 2
**Empirical Novelty And Significance:** 3
**Recommendation:** 3

**Strength And Weaknesses:**

**Strengths**

1. Enforcing node-level alignment in graph mixup tasks is an interesting and convincing effort.
2. The empirical improvements brought by the proposed alignment method seem significant and consistent w.r.t. other mixup strategies.

**Weaknesses**

1. My major concern about this paper is that one of the main technical contributions of this paper, which is the node-level graph matching network, does not seem novel. The proposed graph matching network is simply adapted from Li et al. 2019, adding a softmax-based output after node-feature similarity computation. However, graph matching has been a research topic lasting for decades, and there exists both traditional algorithms and learning-based networks for the matching task. The authors seem to have not fully survived the graph-matching literature, and I believe there exist many papers that offer well-developed solutions for the alignment task in graph mixup.

    For example, I suggest the authors try traditional graph matching methods, for example, RRWM (Cho et al. 2020), IPFP (Leordeanu et al. 2009), or FGM (Zhou et al. 2016), if the previous two methods cannot handle your problem because your graphs are too large. There are also recent deep graph matching neural networks (support node-matching) that are better motivated than the proposed approach, for example, nearly all deep graph matching networks use the Sinkhorn method instead of the proposed softmax method for normalization. The following two deep graph matching papers are suggested for the authors: (Wang et al. 2022) and (Fey et al. 2020).

2. Some graph mixup methods are introduced and discussed in the related work, but they seem to be not compared in experiments, such as SubMix, ifMixUp.  Can the authors explain why they are not compared?

References:
1. Li et al. Graph Matching Networks for Learning the Similarity of Graph Structured Objects. ICML 2019.
1. Cho et al. Reweighted random walks for graph matching. ECCV 2010.
1. Leordeanu et al. An integer projected fixed point method for graph matching and map inference. NIPS 2009.
1. Zhou et al. Factorized Graph Matching. PAMI 2016.
1. Wang et al. Neural Graph Matching Network: Learning Lawler's Quadratic Assignment Problem with Extension to Hypergraph and Multiple-graph Matching. PAMI 2022.
1. Fey et al. Deep Graph Matching Consensus. ICLR 2020.

**Summary Of The Paper:**

This paper presents a new way of graph mixup by enforcing the mixed-up graphs to be softly aligned, via a node-level graph matching network. The graph matching network is based on graph-level similarity learning supervision, whereby its node-level features are utilized to compute the node-wise similarity, followed by a softmax normalization. The authors conduct extensive experiments with different datasets and different mix-up methods, and the improvements brought by the graph alignment step seem convincing

**Summary Of The Review:**

This paper presents a graph node alignment-based graph mixup method, and the empirical results seem to be better than non-aligned graph mixup methods. The graph node matching network, which is one of the major technical contributions of this paper, does not seem novel considering the existence of both learning-free traditional graph matching solvers and learning-based graph matching networks that supports node matching. I suggest the authors to further survey the graph matching literature and improve the technical contributions of this paper.

---

> ### Author Response · Authors · 2022-11-18
> **Responses to reviewer t5LQ part 2/2**
>
> >**Q3: Nearly all deep graph matching networks use the Sinkhorn method instead of the proposed softmax method for normalization.**
>
> Thanks for this constructive comment. We conduct extra experiments to compare the softmax and Sinkhorn normalization. The experiment results in the following tables (Table 5 in the revised manuscript) show that sinkhorn normalization has a similar performance to softmax normalization. Since Sinkhorn is inherently more inefficient than softmax, we follow [3] to use softmax normalization in our framework for efficiency.
>
> |GCN|IMDB-B|PROTEINS|NCI1|REDDIT-B|IMDB-M|REDDIT-M5|
> |----|----|----|----|----|----|----|
> |sinkhorn|73.20 $\pm$ 6.13 | 72.80 $\pm$ 3.72 |  74.89 $\pm$ 1.34 |**90.50 $\pm$ 1.80**| 50.40 $\pm$ 2.83 | 53.13 $\pm$ 2.33 |
> |softmax| **74.40 $\pm$ 5.44** |**73.05 $\pm$ 2.81** | **75.47 $\pm$ 1.49** | 89.30 $\pm$ 2.69 | **50.73 $\pm$ 3.66** | **53.29 $\pm$ 1.97** |
>
> |GIN|IMDB-B|PROTEINS|NCI1|REDDIT-B|IMDB-M|REDDIT-M5|
> |----|----|----|----|----|----|----|
> |sinkhorn|73.10 $\pm$ 6.62 |69.09 $\pm$ 2.84 | 79.95 $\pm$ 2.15 | **91.25 $\pm$ 1.70** | **50.23 $\pm$ 4.70** | 55.09 $\pm$ 1.54 |
> |softmax|**73.40 $\pm$ 6.26** |**69.37 $\pm$ 2.86**|**80.02 $\pm$ 2.45**| 90.55 $\pm$ 2.11| 50.13 $\pm$ 4.34 | **55.19 $\pm$ 1.99**|
>
>
> >**Q4: Some graph mixup methods are introduced but not compared in experiments.**
>
> Thanks for this comment. We have added one more baseline SubMix in our experiments. We have updated Table 2, Table 3, and Figure 3 in the revised manuscript. The results show that our S-Mixup outperforms SubMix on six real-world datasets. In addition, our S-Mixup can improve the robustness of GNNs against corrupted labels better than SubMix.
>
> For the other graph mixup methods, we don't compare them with our methods for the following reasons.
>
> - For ifMixup, it uses an arbitrary node order to align two graphs and
> linearly interpolates adjacency matrices and feature matrices to generate new graph data. As we discussed in Section 5.1, random alignment generates noisy graph data, thereby we don't compare our method with it.
> - Graph Transplant mixes random subgraphs of different input graphs, which is similar to SubMix. Besides, the authors don't provide their implementation. Thus, we believe it is enough to compare with SubMix only.
>
> >Reference
>
> [1] Cho et al. "Reweighted random walks for graph matching." ECCV, 2010.
>
> [2] Wang et al. "Neural Graph Matching Network: Learning Lawler's Quadratic Assignment Problem with Extension to Hypergraph and Multiple-graph Matching." PAMI, 2022.
>
> [3] Fey et al. "Deep Graph Matching Consensus." ICLR, 2020.
>
> [4] Li et al. "Graph matching networks for learning the similarity of graph structured objects." ICML, 2019.

---

> > ### Comment · Reviewer_t5LQ · 2022-11-22
> > **Response to rebuttal**
> >
> > > We would like to point out that the key contribution of our paper is not to solve how to compute the node-level correspondence between graphs.
> >
> > Let me first summarize the technical contributions of this paper:
> > * Introducing the alignment module for graph mix-up, which I agree is novel and interesting.
> > * The node-level matching step, which I believe is incremental, given so many existing graph matching papers that are not cited in the original submission.
> >
> > I agree with the authors that the key contribution is the first point, yet the second point being invalid means that the overall technical quality of this paper might be limited.
> >
> > > As we mentioned in the last sentence of Section 3, in our framework, fast computation of accurate alignments is a prerequisite. Thus, not all of these graph alignment algorithms are appropriate for our framework.
> >
> > I believe the detailed configurations of graph matching methods could be further explored, and it might be too arbitrary to conclude that the softmax-matching approach proposed in your original submission is the best in such a short rebuttal period. For example,
> > * The authors conclude that RRWM is too slow. However, please note that the statistics are based on computers from more than ten years ago. Nowadays, RRWM can even run on GPU.
> > * I also suggest the authors with the FGM method (Zhou et al. 2016) which is more scalable than RRWM. Unfortunately, FGM is overlooked in the rebuttal.
> > * I agree with the authors that obtaining node-level supervision is challenging, however, as also suggested by Reviewer pxKa, self-supervised learning tricks could be applied.
> >
> > > For ifMixup, it uses an arbitrary node order to align two graphs and linearly interpolates adjacency matrices and feature matrices to generate new graph data. As we discussed in Section 5.1, random alignment generates noisy graph data, thereby we don't compare our method with it.
> >
> > Sorry but I still do not get the point of why ifMixup cannot be compared. Is the comparison unfair or something?
> >
> > I am still concerned about the novelty issue after reading the rebuttal, and I believe this issue could not be easily fixed. I intend to retain my score.

---

> ### Author Response · Authors · 2022-11-18
> **Responses to reviewer t5LQ part 1/2**
>
> Thanks for the constructive comments. We have revised the manuscript accordingly and provided pointwise responses below.
>
> >**Q1: My major concern about this paper is that one of the main technical contributions of this paper, which is the node-level graph matching network, does not seem novel.**
>
> We would like to point out that the key contribution of our paper is not to solve how to compute the node-level correspondence between graphs. Instead, we study graph data augmentation by mixup, which is not trivial. The key challenge lies in the fact that **different graphs typically have different numbers of nodes**. Even for graphs with the same number of nodes, there **lacks a node-level correspondence that is required to perform mixup**. While previous graph mixup methods use various tricks to sidestep this problem, we are the first to **explicitly model the node-level correspondence when performing mixup on graphs**. The success of our proposed S-Mixup is based on the well-developed graph alignment methods in the graph matching domain.
>
> >**Q2: I believe there exist many papers that offer well-developed solutions for the alignment task in graph mixup. For example, I suggest the authors try traditional graph matching methods. There are also recent deep graph matching neural networks.**
>
> We agree that there exist many well-developed graph alignment methods in the graph matching domain. As we mentioned in the last sentence of Section 3, in our framework, fast computation of accurate alignments is a prerequisite. Thus, not all of these graph alignment algorithms are appropriate for our framework.
>
> **For traditional graph matching algorithms, it usually takes a long time to compute the alignment**, thereby we don't use traditional graph matching algorithms in our framework. Take the paper you mentioned as an example, RRWM [1] proposes to simulate random walks with reweighting jumping on the association graph. RRWM iteratively updates the solution to achieve robust graph matching. As shown in Figure 2 (a) of their paper, it takes RRWM around 2 seconds to match two graphs with 40 nodes. Such a high computational cost is not suitable for our framework.
>
> Most deep learning based graph matching methods use ground truth correspondences to supervise the training of the networks. For the papers [2][3] you mentioned, both of them use ground truth alignment to train the graph matching neural networks. However, **in our problem, there is a lack of such ground truth correspondences.** We only have the label information of graphs. Thus, we follow [4] to use the triplet loss in Equation (9) to train the graph matching network by incorporating the prior knowledge that the learned representations of graphs from the same class should be more similar than those from different classes.
>
> More importantly, our proposed framework **encodes node-level correspondences when performing graph mixup and is not limited to using the graph matching network proposed in [4].** Other unsupervised learning-based graph matching methods can be easily adopted in our framework. We have added a section to discuss the existing graph matching methods in Appendix D.1 of the revised manuscript.

---

### Official Review · Reviewer_CYp4 · 2022-11-04

**Confidence:** 2
**Correctness:** 3
**Technical Novelty And Significance:** 2
**Empirical Novelty And Significance:** 3
**Recommendation:** 5

**Clarity, Quality, Novelty And Reproducibility:**

There are some sentences not clearly explained to me:
1. In the abstract, can authors explain what is “... any pair of graphs can be mixed directly to generate an augmented graph”?
2. In the introduction, what is “...  and perform mixup as in the case of images”?
3. The last sentence in te introduction can be polished up.


**Strength And Weaknesses:**

## Strengths
1. The problem is well-motivated. Indeed, graph data is highly structured, and the standard augmentation methods can easily destroy the key substructures, which can lead to unexpected biases.
2. The method is quite simple and straightforward. It first applies a graph pair-wise attention to learn a soft alignment score matrix. Then it utilizes this matrix to map graph 2 to the space of graph 1 for alignment.


## Weaknesses
1. Some key notations can be polished up. For example, in eq(5), authors are using A_2 to mix A_1, so the notations can be changed to $X_1’, A_1’, y_1’$.
2. Another question I want to confirm with the authors is, how this S-Mixup can be better than previous methods? I am aware that intuitively, it makes sense, but empirically, I think more qualitative results are required. Also in table 1, why S-Mixup can preserve motif? Can authors also help explain this further?
3. The logic of the placement of Sec 5 is a little confusing to me. Why do the authors want to study the node-level correspondence and graph transformation problems? They seem to jump out without any introduction. Besides, though the conclusion of graph transformation seems correct to me, but I don’t quite follow on why only the graph transformation is studied here? (the first sentence in Sec 5.2) A more common case is when the graph pairs have different number of nodes right?
4. Sec 5.1 and 6.4 can be merged.


**Summary Of The Paper:**

This paper introduces S-Mixup. It is a graph Mixup method, that can better utilize the node-level information between the pair-wise graph.


**Summary Of The Review:**

I’m not an expert on graph with muxip. So I may leave the comments on this point (baselines and experiments) to the other reviewers. The other comments are put above.

---

> ### Author Response · Authors · 2022-11-18
> **Responses to reviewer CYp4**
>
> Thanks for the constructive comments. We have revised the manuscript accordingly and provided pointwise responses below.
>
> >**Q1. How this S-Mixup can be better than previous methods?**
>
> As we mentioned in Section 1, compared to the previous methods, **S-Mixup explicitly models the node-level correspondence to mixup graphs.** In other words, the node-level correspondence between graph pairs in graph mixup is the key to the success of our proposed S-Mixup. Thus, in section 5.1, we use the MOTIF dataset to show the importance of the node-level correspondence. Specifically, we randomly align graph pairs and linearly interpolate node feature matrices and adjacency matrices to generate new graphs.
> **We compare the performance of GIN models trained with the original data and the augmented data.** A GIN model trained on the original data achieves 91.47% accuracy, while the accuracy of the same GIN model trained on the augmented data drops to 52.88%. The significant performance drop clearly demonstrates the importance of node-level correspondence between graphs when mixing graphs.
>
>
> >**Q2. In table 1, why S-Mixup can preserve motifs?**
>
> For graphs in the same class, the same motif exists in them, and an optimal match should align motifs between graph pairs. Thus, the motifs will be preserved if we first align the graph based on the optimal match and then interpolate adjacency matrices and node feature matrices. For example, in Figure 2(a), there should be a one-to-one correspondence between red nodes in the two graphs. If we don't consider such correspondence during graph mixup, we may generate a graph as shown in Figure 2(b). The connections between the red nodes changes and the five red nodes no longer form a cycle motif. If we use the correspondence to mix graphs, then the connections between the red nodes will not change and the five red nodes still form a cycle motif.
> In S-Mixup, we use well-developed graph matching networks to compute accurate alignments between graph pairs, leading to preserving motifs in the mixed graphs.
>
> >**Q3. The logic of the placement of Sec 5 is a little confusing to me. Why do the authors want to study node-level correspondence? It seems to jump out without any introduction.**
>
> As we mentioned in Section 1 and Section 4, our proposed **S-Mixup is motivated by the importance of node-level correspondence between graph pairs** in graph mixup. Thus, in Section 5.1, we show that performing graph mixup without considering node correspondence will degrade the performance of the model. In other words, node-level correspondence matters in graph mixup and we should explicitly model it when performing mixup.
>
> >**Q4. Why only the graph transformation is studied here? A more common case is when the graph pairs have different numbers of nodes right?**
>
> We believe there is a misunderstanding here. **We use the graph transformation to deal with both node number inconsistent and node order missing problems.** In other words, the graph transformation not only makes the graph pairs have the same node order but also makes them have the same number of nodes. After the transformation in Equation (4), the transformed graph $G'_2 = (A'_2, X'_2)$ has the same number of nodes and the same node order as $G_1$. Since this transformation is not perfect, we study the limitation caused by the graph transformation in Section 5.2.
>
> >**Q5. Sec 5.1 and 6.4 can be merged.**
>
> In Section 5.1, we present a case visualization and the performance of the GIN model on the MOTIF dataset, demonstrating the importance of node-level correspondence between graphs when performing mixup. In Section 6.4 (now Appendix F), we would like to show what S-Mixup is doing. Specifically, we not only visualize a case of the mixed graph generated by S-Mixup but also show the relationship between the mixup radio and the normalized GED discussed in Section 5.2. We believe these two sections are talking about **different** things and should not be merged.
>
>
> >**Q6. Clarification of sentences in the introduction.**
>
> - What is “... any pair of graphs can be mixed directly to generate an augmented graph”?
>
>  As we mentioned in Section 4, some existing graph mixup methods don't directly mix two graphs. For example, G-mixup mixes the graph generators of different classes instead of mixing two graph instances. In contrast, given a pair of graphs, our proposed S-Mixup generates new graph data by performing mixup as Equation (4) and (5).
>
> - What is “... and perform mixup as in the case of images”?
>
>  As we mentioned in Section 2.2, the mixup on images linearly interpolates random pairs of data samples and their corresponding labels. In this sentence, we mean linearly interpolating the data samples as Equation (3).
> See more discussions in section 5.

---

### Decision · Program_Chairs · 2023-01-20

**Decision:**

Reject

**Justification For Why Not Higher Score:**

Considering other papers in my batch and the comments of the reviewers, especially the polarizing negative ones, I think some work is still necessary before accepting this work. In particular for what concerns novelty.

**Justification For Why Not Lower Score:**

N/A

**Metareview: Summary, Strengths And Weaknesses:**

This paper presents a data augmentation method for graphs by mixup. It enforces the mixed-up graphs to be softly aligned, via a node-level graph matching network to create a convex combination of a pair of inputs. Extensive experiments are conducted on different datasets and comparing different mix-up methods, the proposed approach seems to consistently improve performance.
I personally believe that more work on data augmentation for graph structured data is needed and this work definitely goes in the right direction. It seems, unfortunately, that few comparisons are simply left out -- i.e. ifMixup -- and that the claim on novelty regarding the soft alignment triggered some concerns with the reviewers. The contributions need to be clearly defined and all the relevant prior art should be included whenever necessary, and it seems reviewers pointed at this problem with the current manuscript. If the matching step is not novel then the overall contribution weakens a lot in my opinion.
For the above reasons I am afraid I have to ask the authors to strengthen their contribution and to resubmit.